# Artistic Style Transfer with Internal-external Learning and Contrastive Learning

**Haibo Chen**    **Lei Zhao**[*]    **Zhizhong Wang**    **Huiming Zhang**

**Zhiwen Zuo**    **Ailin Li**    **Wei Xing**[*]    **Dongming Lu**

College of Computer Science and Technology, Zhejiang University
{cshbchen, cszhl, endywon, qinglanwuji, zzwcs, liailin, wxing, ldm}@zju.edu.cn

## Abstract

Although existing artistic style transfer methods have achieved significant improvement with deep neural networks, they still suffer from artifacts such as disharmonious colors and repetitive patterns. Motivated by this, we propose an internal-external style transfer method with two contrastive losses. Specifically, we utilize internal statistics of a single style image to determine the colors and texture patterns of the stylized image, and in the meantime, we leverage the external information of the large-scale style dataset to learn the human-aware style information, which makes the color distributions and texture patterns in the stylized image more reasonable and harmonious. In addition, we argue that existing style transfer methods only consider the content-to-stylization and style-to-stylization relations, neglecting the stylization-to-stylization relations. To address this issue, we introduce two contrastive losses, which pull the multiple stylization embeddings closer to each other when they share the same content or style, but push far away otherwise. We conduct extensive experiments, showing that our proposed method can not only produce visually more harmonious and satisfying artistic images, but also promote the stability and consistency of rendered video clips.

## 1 Introduction

Artistic style transfer is a long-standing research topic that seeks to render a photograph with a given artwork style. Ever since Gatys *et al.* [10] for the first time proposed a neural method, which leverages a pre-trained Deep Convolutional Neural Network (DCNN) to separate and recombine contents and styles of arbitrary images, an unprecedented booming [20, 26, 15, 30, 36, 51, 48] in style transfer has been witnessed.

Despite the recent progress, there still exists a large gap between real artworks and synthesized stylizations. As shown in Figure 1, the stylized images usually contain some disharmonious colors and repetitive patterns, which makes them easily distinguishable from real artworks. We argue that this is because existing style transfer methods often confine themselves to the internal style statistics of a single artistic image. In some other tasks (for example, image-to-image translation [17, 60, 16, 25, 8, 18]), the style is usually learned from a collection of images, which inspires us to leverage the external information reserved in the large-scale style dataset to improve the stylization results in style transfer. *Why is the external information so important for style transfer?* Our analyses are as follows:

Although different images in the style dataset vary greatly in fine details, they share a key commonality: they are all human-created artworks, whose brushstrokes, color distributions, texture patterns, tones,

---

[*]Corresponding author

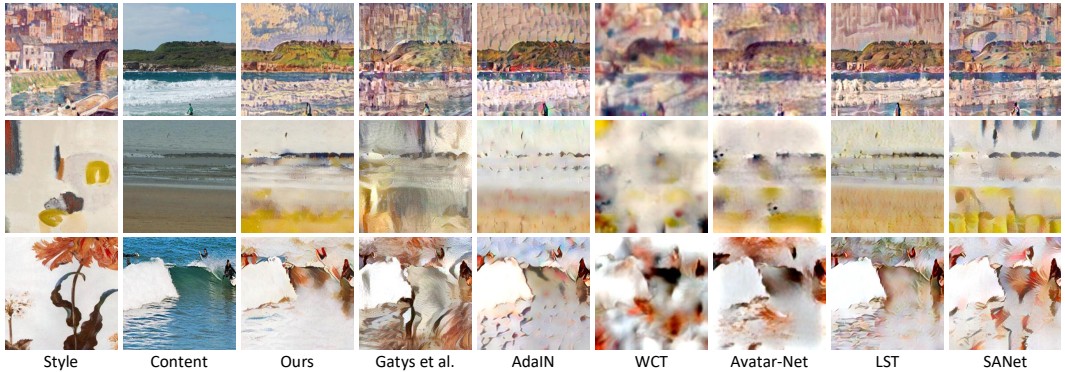

| Style | Content | Ours | Gatys et al. | AdaIN | WCT | Avatar-Net | LST | SANet |

Figure 1: Stylization examples. The first and second columns show the style and content images, respectively. The other seven columns show the stylized images produced by our method, Gatys *et al.* [10], AdaIN [15], WCT [30], Avatar-Net [41], LST [28], and SANet [36].

*etc.*, are more consistent with human perception. Namely, they contain some human-aware style information that is lacked in synthesized stylizations. A natural idea is to utilize such human-aware style information to improve stylization results. To this end, we employ an internal-external learning scheme during training, which takes both internal learning and external learning into consideration. To be more specific, on the one hand, we follow previous methods [10, 20, 46, 54, 58], utilizing internal statistics of a single artwork to determine the colors and texture patterns of the stylized image. On the other hand, we employ Generative Adversarial Nets (GANs) [11, 39, 2, 56, 3] to externally learn the human-aware style information from the large-scale style dataset, which is then used to make the color distributions and texture patterns in the stylized image more reasonable and harmonious, significantly bridging the gap between human-created artworks and AI-created artworks.

In addition, there is another problem with existing style transfer methods: they usually employ a content loss and a style loss to enforce the content-to-stylization relations and style-to-stylization relations, respectively, while neglect the stylization-to-stylization relations, which are also important for style transfer. *What are stylization-to-stylization relations?* Intuitively, stylized images rendered with the same style image should have closer relations in style than those rendered with different style images. Similarly, stylized images based on the same content image should have closer relations in content than those based on different content images. Inspired by this, in this paper we introduce two contrastive losses: content contrastive loss and style contrastive loss that can pull the multiple stylization embeddings closer to each other when they share the same content or style, but push far away otherwise. To the best of our knowledge, this is the first work that successfully leverages the power of contrastive learning [6, 12, 21, 38] in the style transfer scenario.

Our extensive experiments show that the proposed method can not only produce visually more harmonious and plausible artistic images, but also promote the stability and consistency of rendered video clips.

To summarize, the main contributions of this work are threefold:

- We propose a novel internal-external style transfer method which takes both internal learning and external learning into consideration, significantly bridging the gap between human-created and AI-created artworks.
- We for the first time introduce contrastive learning to style transfer, yielding more satisfying stylization results with the learned stylization-to-stylization relations.
- We demonstrate the effectiveness and superiority of our approach by extensive comparisons with several state-of-the-art artistic style transfer methods.

## 2   Related Work

**Artistic style transfer.** Artistic style transfer is an image editing task that aims at transferring artistic styles onto everyday photographs to create new artworks. Earlier methods usually resort to traditional

techniques such as stroke rendering [13], image analogy [14, 42, 9, 31], and image filtering [52] to perform artistic style transfer. These methods typically rely on low-level statistics and often fail to capture semantic information. Recently, Gatys *et al.* [10] discovered that the Gram matrix upon deep features extracted from a pre-trained DCNN can notably represent the characteristics of visual styles, which opens up the neural style transfer era. Since then, a suite of neural methods have been proposed, boosting the development of style transfer from different concerns. Specifically, [20, 27, 46] utilize feed-forward networks to improve efficiency. [26, 54, 36, 58, 35] refine various elements in the stylized images (including content preservation, textures, brushstrokes, *etc.*) to enhance visual quality. [7, 15, 30, 41, 28] propose universal style transfer methods to achieve generalization. [29, 47, 51] inject random noise to the generative network to encourage diversity. Despite the rapid progress, these style transfer methods still suffer from spurious artifacts such as disharmonious colors and repetitive patterns.

Notice that there is another line of work [40, 24, 23, 45, 4, 5] that aims to learn *an artist's style* from all his/her artworks. In comparison, instead of learning *an artist's style*, we focus on better leaning *an artwork's style* (just like the style transfer methods mentioned in the previous paragraph) with the assist of the human-aware style information reserved in the external style dataset. Therefore, our method is orthogonal to these works.

**Image-to-image translation.** Image-to-image translation (I2I) [17, 60, 16, 25, 8, 18] aims at learning the mapping between different visual domains, which is closely related to style transfer. [60, 16] have distinguished these two tasks: (i) I2I can only translate between content-similar visual domains (such as horses↔zebras and summer↔winter), while style transfer does not have such limitation, whose content image and style image can be totally different (*e.g.*, the former is a photo of a person and the latter is van Gogh's *The Starry Night*). (ii) I2I aims to learn the mapping between two image collections, while style transfer aims to learn the mapping between two specific images. However, we argue that we can borrow some insights from I2I, and leverage the external information of the large-scale style image collections to improve the stylization quality in style transfer.

**Internal-external learning.** Internal-external learning has shown effectiveness in various image generation tasks, such as super-resolution, image inpainting, and so on. In detail, Soh *et al.* [44] presented a fast, flexible, and lightweight self-supervised super-resolution method by exploiting both external and internal samples. Park *et al.* [37] developed an internal-external super-resolution method that facilitates super-resolution networks to further enhance the quality of the restored images. *Wang et al.* [49] proposed a general external-internal learning inpainting scheme, which learns semantic knowledge externally by training on large datasets while fully utilizes internal statistics of the single test image. However, in the field of style transfer, existing methods only use a single artistic image to learn style, resulting in unsatisfying stylization results. Motivated by this, in this work we propose an internal-external style transfer method that takes both internal learning and external learning into consideration, significantly bridging the gap between human-created and AI-created artworks.

**Contrastive learning.** Generally, there are three key ingredients in a contrastive learning process: query, positive examples, and negative examples. The target of contrastive learning is to associate a "query" with its "positive" example while disassociate the "query" with other examples that are referred to as "negatives". Recently, contrastive learning has demonstrated its effectiveness in the field of conditional image synthesis. To be more specific, ContraGAN [21] introduced a conditional contrastive loss (2C loss) to learn both data-to-class and data-to-data relations. Park *et al.* [38] maximized the mutual information between input and output with contrastive learning to encourage content preservation in unpaired image translation problems. Liu *et al.* [34] introduced a latent-augmented contrastive loss to encourage images generated from adjacent latent codes to be similar and those generated from distinct latent codes to be dissimilar, achieving diverse image synthesis. Yu *et al.* [55] proposed a dual contrastive loss in adversarial training that generalizes representation to more effectively distinguish between real and fake, and further incentivizes the image generation quality. Wu *et al.* [53] improved the image dehazing result by introducing contrastive learning, which ensures that the restored image is pulled closer to the clear image and pushed far away from the hazy image in representation space.

Note that all the above contrastive learning methods cannot be adopted for style transfer. In this work, we make the first attempt to adapt contrastive learning to artistic style transfer, and propose two novel contrastive losses: content contrastive loss and style contrastive loss to learn the stylization-to-stylization relations that are ignored by existing style transfer methods.

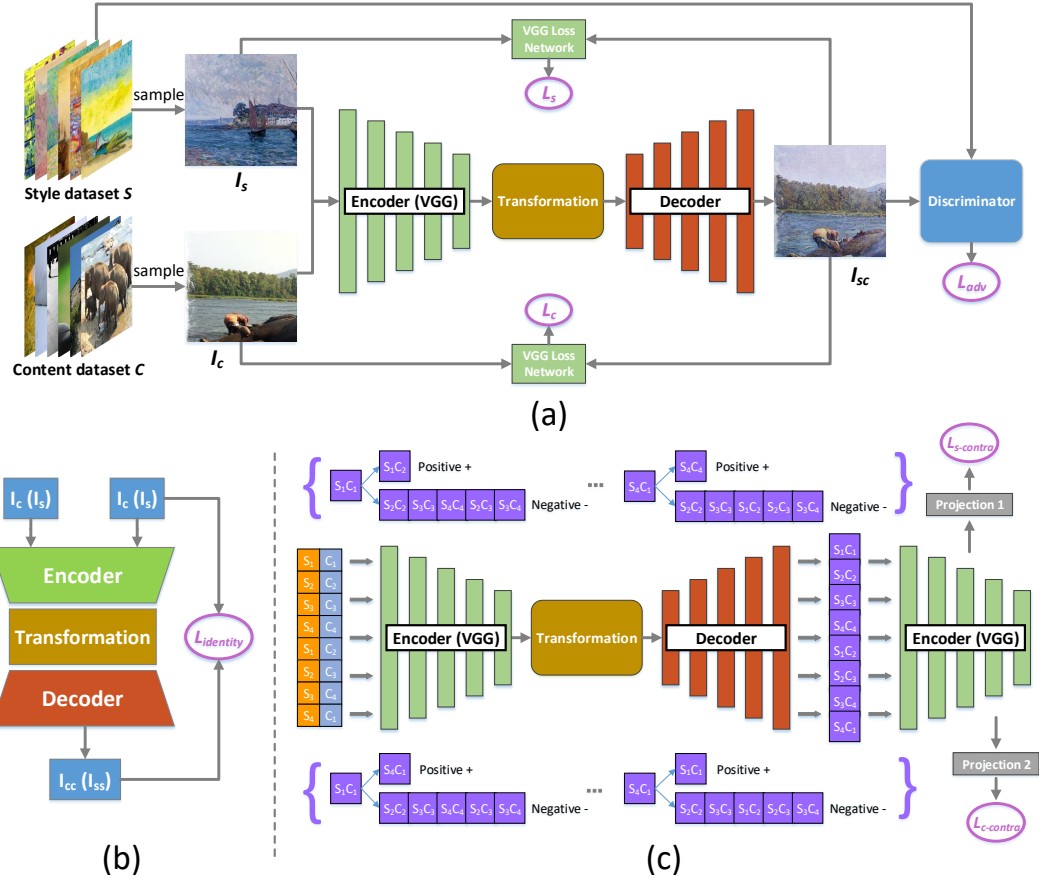

Figure 2: An overview of the proposed method. (a) illustrates our basic framework, which mainly contains a pre-trained encoder, a style-attentional transformation module, a decoder, and a discriminator. The style loss $\mathcal{L}_s$ and the content loss $\mathcal{L}_c$ are used to learn the style and content information, respectively. The adversarial loss $\mathcal{L}_{adv}$ is used to learn the human-aware style information. (b) and (c) depict the identity loss $\mathcal{L}_{identity}$ and contrastive losses $\mathcal{L}_{s-contra}$ & $\mathcal{L}_{c-contra}$, where $\mathcal{L}_{identity}$ is used to preserve more content structures and style characteristics in the stylized image, and $\mathcal{L}_{s-contra}$ & $\mathcal{L}_{c-contra}$ are used to learn the stylization-to-stylization relations.

## 3  Proposed Method

Existing style transfer methods usually produce unsatisfying stylization results with disharmonious colors and repetitive patterns, which makes them pretty easy to be distinguished from real artworks. As an attempt to bridge the large gap between human-created and AI-created artworks, we propose a novel internal-external style transfer method with two contrastive losses. The overview of our method is shown in Figure 2. It is worth noting that our framework is built on the SANet [36] (one of the state-of-the-art style transfer methods) backbone, which consists of an encoder $E$, a transformation module $T$, and a decoder $D$. In detail, $E$ is a pre-trained VGG-19 network [43] used to extract image features, $T$ is a style-attentional network that can flexibly match the semantic nearest style features onto the content features, and $D$ is a generative network used to transform encoded semantic feature maps into stylized images. We extend SANet [36] with our proposed changes, and our full model is described below.

### 3.1  Internal-external Learning

Let $C$ and $S$ be the sets of photographs and artworks, respectively. We aim to learn both the internal style characteristics from a single artwork $I_s \in S$ and the external human-aware style information

from the dataset $S$, and then transfer them to an arbitrary content image $I_c \in C$ to create new artistic images $I_{sc}$.

**Internal style learning.** Following previous style transfer methods [15, 36, 1], we use a pre-trained VGG-19 network $\phi$ to capture the internal style characteristics from a single artistic image, and the style loss can be generally computed as:

$$\mathcal{L}_s := \sum_{i=1}^{L} \| \mu(\phi_i(I_{sc})) - \mu(\phi_i(I_s)) \|_2 + \| \sigma(\phi_i(I_{sc})) - \sigma(\phi_i(I_s)) \|_2 \tag{1}$$

where $\phi_i$ denotes the $i_{th}$ layer (Relu1_1, Relu2_1, Relu3_1, Relu4_1, and Relu5_1 layers are used in our model) of the VGG-19 network. $\mu$ and $\sigma$ represent the mean and standard deviation of feature maps extracted by $\phi_i$, respectively.

**External style learning.** Here, we employ GAN [11, 39, 2, 56, 3] to learn the human-aware style information from the style dataset $S$. GAN is a popular generative model consisting of two networks (*i.e.*, a generator $\mathcal{G}$ and a discriminator $\mathcal{D}$) that compete against each other. Specifically, we input the stylized images produced by the generator and the artworks sampled from $S$ to the discriminator as fake data and real data, respectively. In the training process, the generator will try to fool the discriminator by generating a realistic artistic image, while the discriminator will try to distinguish generated fake artworks from real ones. Joint training of these two networks leads to a generator that is able to produce remarkable realistic fake images with the learned human-aware style information. The adversarial training process can be formulated as (note that our generator $\mathcal{G}$ contains an encoder $E$, a transformation module $T$, and a decoder $D$, as shown in Figure 2 (a)):

$$\mathcal{L}_{adv} := \mathop{\mathbb{E}}_{I_s \sim S}[log(\mathcal{D}(I_s))] + \mathop{\mathbb{E}}_{I_c \sim C, I_s \sim S}[log(1 - \mathcal{D}(D(T(E(I_c), E(I_s)))))] \tag{2}$$

**Content structure preservation.** To preserve the content structure of $I_c$ in the stylized image $I_{sc}$, we adopt the widely-used perceptual loss:

$$\mathcal{L}_c := \| \phi_{conv4\_2}(I_{sc}) - \phi_{conv4\_2}(I_c) \|_2 \tag{3}$$

**Identity loss.** Similar to [36, 32, 59], we utilize the identity loss to encourage the generator $\mathcal{G}$ to be an approximate identity mapping when the content image and style image are the same. In this manner, more content structures and style characteristics can be preserved in the stylization result. The identity loss is depicted in Figure 2 (b) and defined as:

$$\mathcal{L}_{identity} := \lambda_{identity1}(\| I_{cc} - I_c \|_2 + \| I_{ss} - I_s \|_2) +$$
$$\lambda_{identity2} \sum_{i=1}^{L} (\| \phi_i(I_{cc}) - \phi_i(I_c) \|_2 + \| \phi_i(I_{ss}) - \phi_i(I_s) \|_2) \tag{4}$$

where $I_{cc}$ is the output image generated when both the content image and style image are $I_c$. $I_{ss}$ is analogous. $\lambda_{identity1}$ and $\lambda_{identity2}$ are the weights associated with different loss terms. For $\phi_i$, we choose Relu1_1, Relu2_1, Relu3_1, Relu4_1, and Relu5_1 layers in our experiments.

### 3.2 Contrastive Learning

Intuitively, stylized images rendered with the same style image should have closer relations in style than those rendered with different style images. Similarly, stylized images based on the same content image should have closer relations in content than those based on different content images. We refer to such relations as stylization-to-stylization relations. Generally, existing style transfer methods only consider the content-to-stylization and style-to-stylization relations by applying the content loss and style loss (like $\mathcal{L}_c$ and $\mathcal{L}_s$ introduced above), while neglect the stylization-to-stylization relations. To tackle this problem, we for the first time introduce contrastive learning to style transfer. The core idea of contrastive learning is to associate data points with their "positive" examples while disassociate them from the other points that are regarded as "negatives".

Specifically, we propose two contrastive losses: a style contrastive loss and a content contrastive loss to learn the stylization-to-stylization relations. Note that for clearer expression, hereafter, we use $s_i$ to represent the $i_{th}$ style image, $c_i$ to represent the $i_{th}$ content image, and $s_i c_i$ to represent the stylized image generated with $s_i$ and $c_i$. To perform contrastive learning in every training batch, we arrange a batch of style and content images in the following manner:

Assume the batch size = $b$, which is an even number. Then we get a batch of style images $\{s_1, s_2, ..., s_{b/2}, s_1, s_2, ..., s_{b/2-1}, s_{b/2}\}$, and a batch of content images $\{c_1, c_2, ..., c_{b/2}, c_2, c_3, ..., c_{b/2}, c_1\}$. Hence, the corresponding stylized images are $\{s_1 c_1, s_2 c_2, ..., s_{b/2} c_{b/2}, s_1 c_2, s_2 c_3, ..., s_{b/2-1} c_{b/2}, s_{b/2} c_1\}$. In this way, we ensure that for every stylized image $s_i c_j$, we can find a stylized image $s_i c_x$ ($x \neq j$) that shares the same style with it, and a stylized image $s_y c_j$ ($y \neq i$) that shares the same content with it in the same batch. Figure 2 (c) depicts this process by taking $b = 8$ as an example.

**Style contrastive loss.** To associate stylized images that share the same style, for a stylized image $s_i c_j$, we select $s_i c_x$ ($x \neq j$) as its positive example ($s_i c_x$ shares the same style with $s_i c_j$), and $s_m c_n$ ($m \neq i$ and $n \neq j$) as its negative examples. Notice that $s_m c_n$ represents a series of stylized images, not just one image. Then we can formulate our style contrastive loss as follows:

$$\mathcal{L}_{s-contra} := -log(\frac{exp(l_s(s_i c_j)^T l_s(s_i c_x)/\tau)}{exp(l_s(s_i c_j)^T l_s(s_i c_x)/\tau) + \sum exp(l_s(s_i c_j)^T l_s(s_m c_n)/\tau)}) \quad (5)$$

where $l_s = h_s(\phi_{relu3\_1}(\cdot))$, in which $h_s$ is a style projection network. $l_s$ is used to obtain the style embeddings from stylized images. $\tau$ is a temperature hyper-parameter to control push and pull force.

**Content contrastive loss.** Similar to the style contrastive loss, to associate stylized images that share the same content, for a stylized image $s_i c_j$, we select $s_y c_j$ ($y \neq i$) as its positive example ($s_y c_j$ shares the same content with $s_i c_j$), and $s_m c_n$ ($m \neq i$ and $n \neq j$) as its negative examples. We express the content contrastive loss as:

$$\mathcal{L}_{c-contra} := -log(\frac{exp(l_c(s_i c_j)^T l_c(s_y c_j)/\tau)}{exp(l_c(s_i c_j)^T l_c(s_y c_j)/\tau) + \sum exp(l_c(s_i c_j)^T l_c(s_m c_n)/\tau)}) \quad (6)$$

where $l_c = h_c(\phi_{relu4\_1}(\cdot))$, in which $h_c$ is a content projection network. $l_c$ is used to obtain the content embeddings from stylized images.

### 3.3 Final Objective

We summarize all aforementioned losses and obtain the final objective of our model,

$$\mathcal{L}_{final} := \lambda_1 \mathcal{L}_s + \lambda_2 \mathcal{L}_{adv} + \lambda_3 \mathcal{L}_c + \lambda_4 \mathcal{L}_{identity} + \lambda_5 \mathcal{L}_{s-contra} + \lambda_6 \mathcal{L}_{c-contra} \quad (7)$$

where $\lambda_1$, $\lambda_2$, $\lambda_3$, $\lambda_4$, $\lambda_5$, and $\lambda_6$ are hyper-parameters for striking proper balance among losses.

## 4 Experimental Results

In this section, we first introduce the experimental settings. Then we present qualitative and quantitative comparisons between the proposed method and several baseline models. Finally, we discuss the effect of each component in our model by conducting ablation studies.

### 4.1 Experimental Settings

**Implementation details.** We build on the recent SANet [36] backbone and extend it with our proposed changes to further push the boundaries in automatic artwork generation. We refer to the original paper [36] for the detailed network architecture of the encoder $E$, transformation module $T$, and decoder $D$. As for the discriminator $\mathcal{D}$, we employ the multi-scale discriminator proposed by Wang *et al.* [50]. The style projection network $h_s$ is a two-layer MLP (Multilayer Perceptron) with 256 units at the first layer and 128 units at the second layer. Similarly, the content projection network $h_c$ is a two-layer MLP with 128 units at each layer. The hyper-parameter $\tau$ in Equation (5) and (6) is set to 0.2. The loss weights in Equation (4) and (7) are set to $\lambda_{identity1} = 50$, $\lambda_{identity2} = 1$, $\lambda_1 = 1$, $\lambda_2 = 5$, $\lambda_3 = 1$, $\lambda_4 = 1$, $\lambda_5 = 0.3$, and $\lambda_6 = 0.3$. We train our network using the Adam optimizer

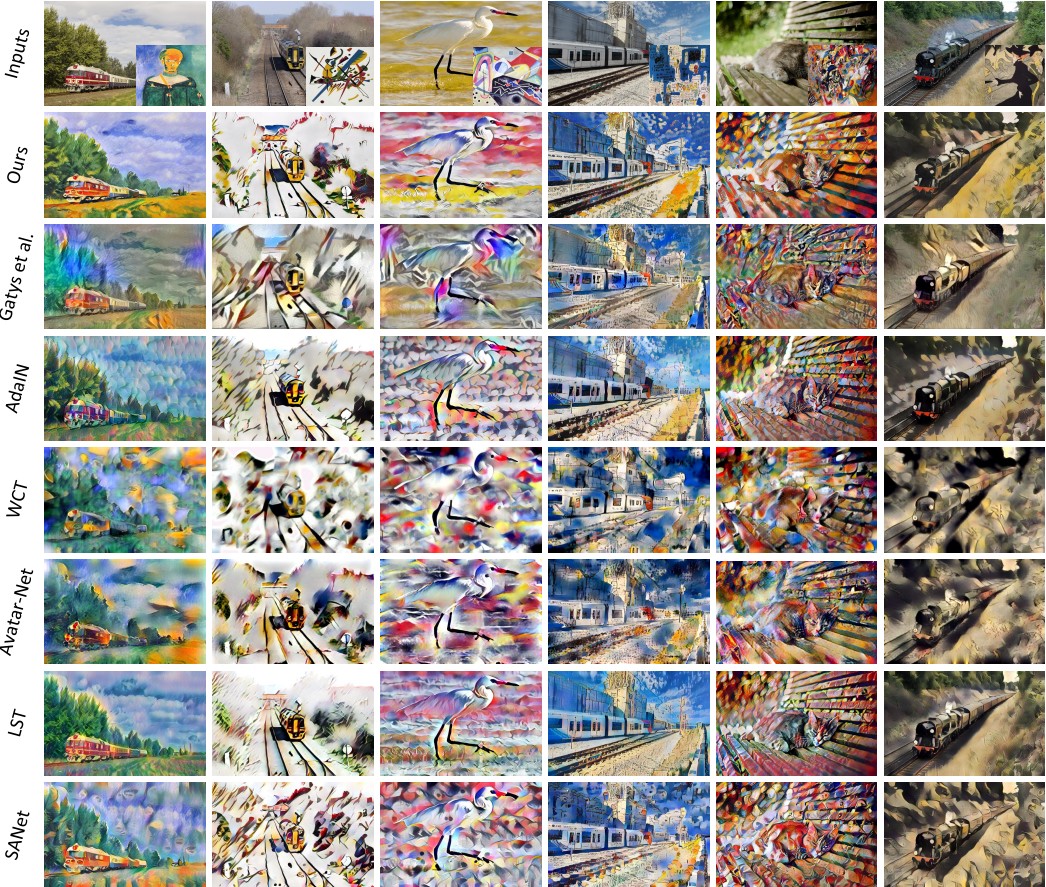

Figure 3: Qualitative comparisons on image style transfer. The first row shows the content and style images. The rest of the rows show the stylization results generated with different style transfer methods.

with a learning rate of 0.0001 and a batch size of 16 for 160000 iterations. Our code is available at: https://github.com/HalbertCH/IEContraAST.

**Datasets.** Like [15, 58, 36, 19], we take MS-COCO [33] and WikiArt [22] as the content dataset and style dataset, respectively. During the training stage, we first resize the smallest dimension of training images to 512 while preserving the aspect ratio, and then randomly crop $256 \times 256$ patches from these images as input. Note that in the reference stage, our method is applicable for content images and style images with any size.

**Baselines.** We choose several state-of-the-art style transfer methods as our baselines, including Gatys *et al.* [10], AdaIN [15], WCT [30], Avatar-Net [41], LST [28], and SANet [36]. All these methods are conducted by using the public codes and default configurations.

### 4.2 Qualitative Comparisons

In Figure 3, we show the qualitative comparisons between our method and six baselines introduced above. We observe that Gatys *et al.* [10] is prone to fall in a bad local minimum (*e.g.*, $1^{st}$, $2^{nd}$, and $3^{rd}$ columns). AdaIN [15] sometimes produces messy stylized images with unseen colors and unwanted halation around the edges (*e.g.*, $1^{st}$, $3^{rd}$, and $6^{th}$ columns). WCT [30] often introduces distorted patterns, yielding less-structured and blunt stylized images (*e.g.*, $2^{nd}$, $4^{th}$, and $5^{th}$ columns). Avatar-Net [41] is hard to produce sharp details and fine brushstrokes (*e.g.*, $1^{st}$, $4^{th}$, and $5^{th}$ columns). LST [28] usually produces less stylized images with very limited texture patterns (*e.g.*, $2^{nd}$, $4^{th}$, and $6^{th}$ columns). SANet [36] tends to apply similar repeated texture patterns among different styles (*e.g.*, $1^{st}$, $3^{rd}$, and $6^{th}$ columns). Despite the recent progress, the gap between synthesized artistic

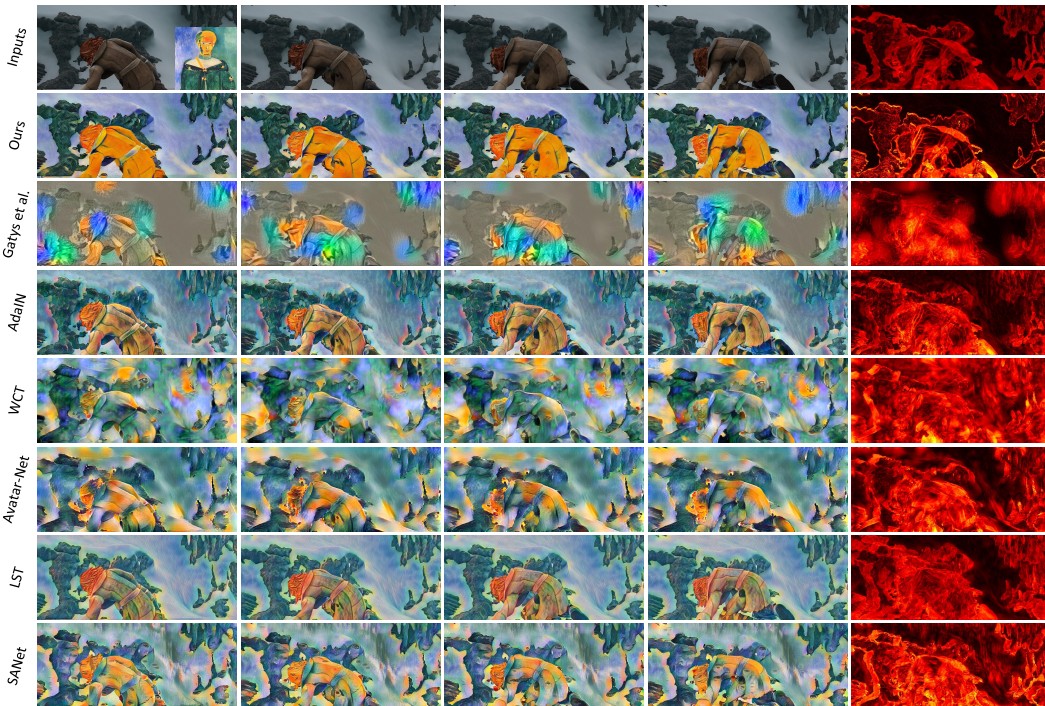

Figure 4: Qualitative comparisons on video style transfer. The first row shows several video frames and the style image. The rest of the rows show the stylization results generated with different style transfer methods. The last column shows the heat maps of differences between different frames.

Table 1: The user study scores for different methods. The higher the better.

|  | WikiArt | Gatys et al. | AdaIN | WCT | Avatar-Net | LST | SANet | Ours |
|---|---|---|---|---|---|---|---|---|
| Preference Score | - | 0.143 | 0.118 | 0.099 | 0.087 | 0.125 | 0.161 | **0.267** |
| Deception Score | 0.875 | 0.438 | 0.363 | 0.375 | 0.275 | 0.381 | 0.394 | **0.624** |

images and real artworks is still very large. To further narrow this gap, we introduce internal-external learning and contrastive learning to artistic style transfer, leading to visually more harmonious and plausible artistic images, as shown in the $2^{nd}$ row of Figure 3.

We also compare our method with 6 baselines on video style transfer, which is conducted between a content video and a style image in a frame-wise manner. The stylization results are shown in Figure 4. To visualize the stability and consistency of synthesized video clip, we also show the heat maps of differences between different frames in the last column of Figure 4. As we can see, our approach outperforms existing style transfer methods in terms of stability and consistency by a significant margin. This can be attributed to two points: (i) external learning smooths the stylization results by eliminating those distorted texture patterns; (ii) the proposed contrastive losses take the stylization-to-stylization relations into consideration, pulling adjacent stylized frames closer to each other since they share the same style and similar content.

## 4.3 Quantitative Comparisons

As the qualitative assessment presented above could be subjective, in this section, we resort to several evaluation metrics to better assess the performance of the proposed method in a quantitative manner.

User study [54, 36, 24, 23, 48] is the most widely adopted evaluation metric in style transfer, which investigates user preference over different stylization results for a more objective comparison.

**Preference score.** We use 10 content images and 15 style images to synthesize 150 stylized images for each method. Then 20 content-style pairs are randomly selected for each participant and show

Table 2: The average LPIPS distances for different methods. The lower the better.

| | Inputs | Gatys et al. | AdaIN | WCT | Avatar-Net | LST | SANet | Ours |
|---|---|---|---|---|---|---|---|---|
| LPIPS Distance | 0.231 | 0.488 | 0.369 | 0.460 | 0.341 | 0.326 | 0.372 | **0.317** |

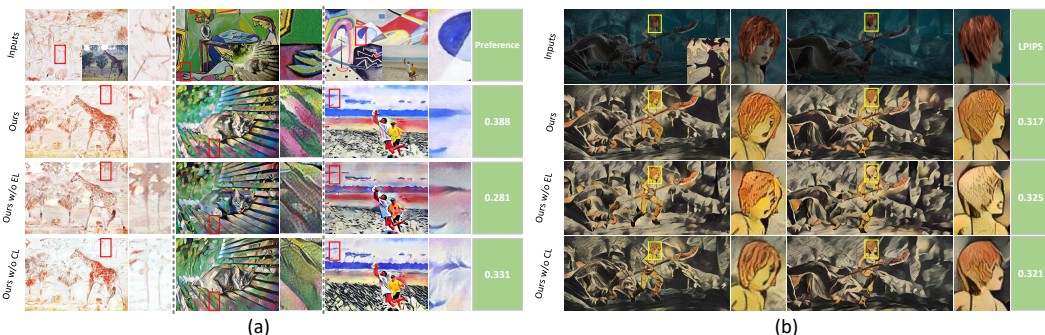

Figure 5: Ablation studies of external learning (*abbr.* EL) and contrastive learning (*abbr.* CL) on (a) image style transfer and (b) video style transfer. Please zoom in for a better view and details.

them the stylized images generated by our and competing methods side-by-side in a random order. Next, we ask each participant to choose his/her favorite stylization result for each content-style pair. We finally collect 1000 votes from 50 participants and present the percentage of votes for each method in the second row of Table 1. The results indicate that the stylized images generated by our method are more preferred by human participants compared to those generated by the competing methods.

**Deception score.** To measure the gap between AI-created artistic images and human-created artworks, we conduct another user study: for each participant, we show them 80 artistic images which consist of 10 human-created artworks collected from WikiArt [22] and 70 stylized images generated by our and 6 baseline methods (note that each method provides 10 stylized images). Then for every image, we ask these participants to guess if it is a real artwork or not. The deception score is calculated as the fraction of times that the stylized images generated by this method are identified as "real". For comparison, we also report the fraction of times that the human-created artworks are identified as "real". The results are shown in the third row of Table 1, where we can see that the deception rate of our method is closest to that of human-created artworks, further demonstrating the effectiveness of our method.

To quantitatively evaluate the stability and consistency of the proposed method on video style transfer, we adopt LPIPS (*Learned Perceptual Image Patch Similarity*) [57] as the evaluation metric.

**LPIPS.** LPIPS is a widely used metric in the field of multimodal image-to-image translation (MI2I) [61, 16, 25, 8] to measure diversity. In this paper, we employ LPIPS to measure the stability and consistency of rendered clips by computing the average perceptual distances between adjacent frames. Note that contrary to MI2I methods that expect a higher LPIPS value to achieve better diversity, we expect a lower LPIPS value to achieve better stability and consistency. We synthesize 18 stylized video clips for each method and report the average LPIPS distances in Table 2, where we observe that our approach obtains the best score among all methods, consistent with the qualitative comparisons in Figure 4.

## 4.4 Ablation Studies

In this section, we conduct several ablation studies to highlight the effect of different components in our model.

We first explore the effect of external learning (*abbr.* EL) and contrastive learning (*abbr.* CL) on image style transfer. As for internal learning, since its effect has been fully validated in existing style transfer methods, we do not ablate it in this experiment. Figure 5 (a) shows the image stylization results of our method with and without EL/CL. It can be observed that, without EL, the stylized images become messier with abrupt colors and obvious distortions. The reason could be that the

model without EL only focuses on increasing the style similarity between the stylized image and the style image, without considering whether the color distributions and texture patterns in the stylized image are natural and harmonious. In comparison, the model with EL can learn the human-aware style information from the large-scale style dataset, leading to more realistic and harmonious stylized images that cannot be distinguished from real artworks by the discriminator. In addition, we also find that our method can better match the target style to the content image with the proposed contrastive losses. This is because our contrastive losses can help the network to learn better style and content representations by taking the stylization-to-stylization relations into consideration, further refining the stylization results. The user preference results reported in the last column of Figure 5 (a) also demonstrate that our full model has the best performance.

Similar ablation studies are also conducted on video style transfer. As shown in Figure 5 (b), the stability degradation can be observed after we remove external learning or contrastive learning from our method (notice the color of hair and skin), which is in line with the reported LPIPS distance. The results indicate that both external learning and contrastive learning can improve the stability of video style transfer. As we analyzed in Section 4.2, external learning obtains stability gains by eliminating distorted texture patterns, and contrastive learning obtains stability gains by pulling adjacent stylized frames closer to each other.

## 5 Limitations

One limitation of this work is that the proposed internal-external learning scheme and two contrastive losses cannot be applied to learning-free style transfer methods, such as WCT [30], Avatar-Net [41], LST [28], *etc.* This is because the training process is necessary for our method. Therefore, our method can only be incorporated into learning-based methods, such as Johnson *et al.* [20], AdaIN [15], SANet [36] (in this work, we mainly take SANet as our backbone to show the effectiveness and superiority of our method), *etc.* Another limitation is that in the inference stage, the style images that are too different from the training styles may not benefit from the external learning scheme, since they are out of the learned style distributions.

## 6 Conclusion

In this paper, we propose an internal-external style transfer method with two novel contrastive losses. The internal-external learning scheme learns simultaneously both the internal statistics from a single artistic image and the human-aware style information from the large-scale style dataset. As for the contrastive losses, they are dedicated to learning the stylization-to-stylization relations by pulling the multiple stylization embeddings closer to each other when they share the same content or style, but pushing far away otherwise. Extensive experiments show that our method can not only produce visually more harmonious and satisfying artistic images, but also significantly promote the stability and consistency of rendered video clips. The proposed method is simple and effective, and may shed light on more future understandings of artistic style transfer from a new perspective. In the future, we would like to extend our method to other vision tasks, for example, texture synthesis.

## Acknowledgments

This work was supported in part by the projects No. 2020YFC1523202, 19ZDA197, LY21F020005, 2021009, 2019C03137, MOE Frontier Science Center for Brain Science & Brain-Machine Integration (Zhejiang University), National Natural Science Foundation of China (Research on Key Technologies of art image restoration based on decoupling learning), and Key Scientific Research Base for Digital Conservation of Cave Temples (Zhejiang University), State Administration for Cultural Heritage. We would also like to thank the reviewers and AC for their constructive and insightful comments on the early submission.

## Funding Transparency Statement

The projects mentioned in our **Acknowledgments** provided funding and support to this work. There are no additional revenues related to this work.

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
