# The Supplementary Material for "Artistic Style Transfer with Internal-external Learning and Contrastive Learning"

**Haibo Chen**    **Lei Zhao**∗    **Zhizhong Wang**    **Huiming Zhang**

**Zhiwen Zuo**    **Ailin Li**    **Wei Xing**∗    **Dongming Lu**

College of Computer Science and Technology, Zhejiang University
{cshbchen, cszhl, endywon, qinglanwuji, zzwcs, liailin, wxing, ldm}@zju.edu.cn

## 1   Societal Impacts

**Positive impacts.** From our point of view, there may be three kinds of people that can benefit from this research: (i) Researchers who are interested in style transfer. Our research may inspire them to develop some more effective and remarkable methods in the future. (ii) Artists. They may draw some inspiration from the artistic images generated by our approach to augment their own work. (iii) Ordinary people who are interested in art. They can create stunning artistic images without any knowledge backgrounds about art.

**Negative impacts.** One potential negative impact is that the proposed style transfer method may replace some human jobs.

## 2   Involved Assets

Existing assets that we used in this work mainly include: 1) the codes of Gatys *et al.* [1], AdaIN [2], WCT [5], Avatar-Net [8], LST [4], and SANet [7], and 2) the MS-COCO dataset [6] and WikiArt dataset [3]. We report their URLs and licenses in the following,

- Gatys *et al.*: https://github.com/anishathalye/neural-style, GNU GENERAL PUBLIC LICENSE v3.0.

- AdaIN: https://github.com/naoto0804/pytorch-AdaIN, MIT License.

- WCT: https://github.com/eridgd/WCT-TF, MIT License.

- Avatar-Net: https://github.com/LucasSheng/avatar-net, we were unable to find its license.

- LST: https://github.com/sunshineatnoon/LinearStyleTransfer, BSD 2-Clause License.

- SANet: https://github.com/GlebBrykin/SANET, MIT License.

- MS-COCO: https://cocodataset.org/#download, we were unable to find its license.

- WikiArt: https://www.kaggle.com/c/painter-by-numbers, we were unable to find its license.

Note that MS-COCO and WikiArt have been widely used in a lot of existing works, and as far as we know, they do not contain personally identifiable information or offensive content.

---

∗Corresponding author

35th Conference on Neural Information Processing Systems (NeurIPS 2021).

# 3   Research with Human Subjects

(i) In this work, we conducted user studies to evaluate the performance of our method. Section 4.3 in our main paper has presented the detailed text of instructions. Here, we provide the corresponding screenshots in Figure 1.

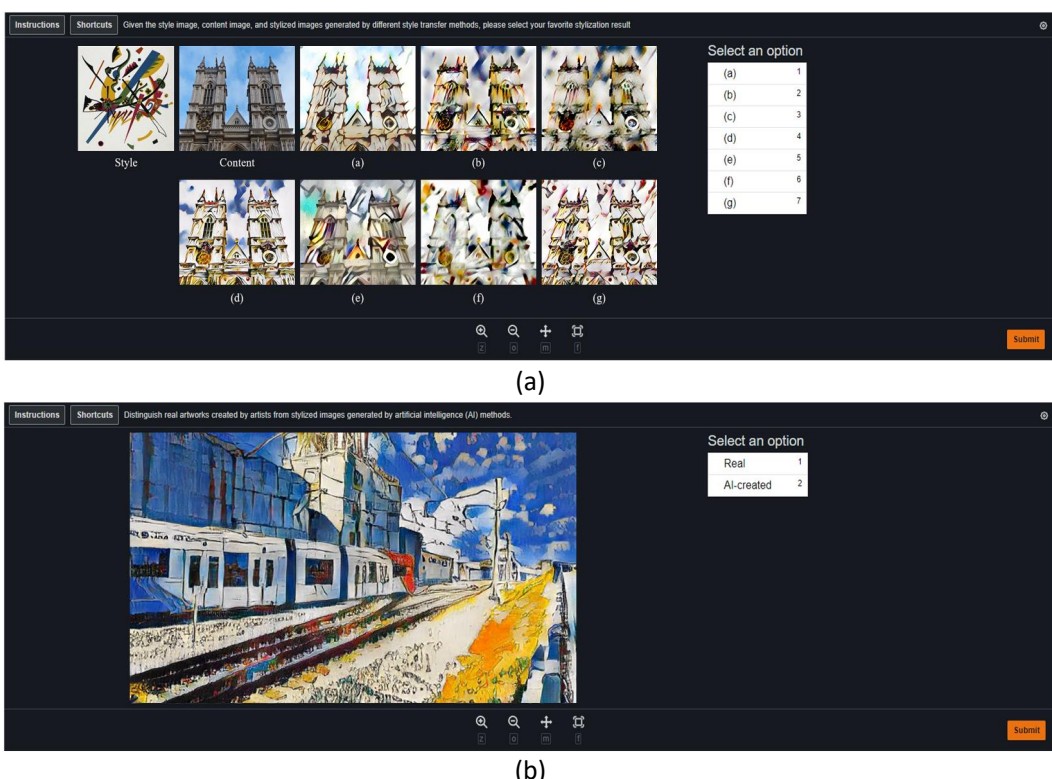

Figure 1: The screenshots of user study in terms of (a) preference score and (b) deception score.

(ii) As far as we know, our user studies do not involve any potential participant risks.

(iii) As for the wage, we paid each participant $0.5 for completing an assignment within 40 minutes. We determined this wage mainly based on our research about other similar tasks that are also published on the Amazon Mechanical Turk (AMT) platform.

# 4   More Experimental Results

In this section, we present more experimental results: (i) We present more ablation study results in Figure 2 to demonstrate the effectiveness of the proposed internal-external learning and contrastive learning scheme. (ii)More qualitative comparisons between our method and the baselines are presented in Figure 3 and Figure 4.

# 5   Resources

During our experiments, 2 GeForce GTX 1080 Ti GPUs and 2 GeForce RTX 3090 GPUs are used.