# OpenReview forum: "Artistic Style Transfer with Internal-external Learning and Contrastive Learning"
_NeurIPS.cc/2021/Conference — NeurIPS 2021 Poster_

### Official Review · Reviewer_vsGU · 2021-07-12

**Rating:** 6
**Confidence:** 3

**Summary:**

The paper describes a method for feed-forward image stylization that combines several losses. The novel losses are a contrastive loss that optimizes the style embedding of the training image to match the embedding of a target style and content and to be distinct from other style and contents.  Other losses used during training include feature mean/variances and a GAN loss.


**Limitations And Societal Impact:**

The paper doesn't discuss limitations.

**Main Review:**

Overall, the visual quality of the results (Figure 3) looks substantially better than the baselines. The detailed structure and scene contours are preserved better than the baselines, and the regions are filled with sensible textures from the exemplars. I haven't attempted to survey the literature to judge if these are fair representations of the baselines

It's unclear to me how these improvements results from the contrastive losses. Better representation of image structure doesn't seem to encoded in the contrastive losses.

The writing of the paper is very difficult to follow and I'm not sure that it's reproducible. Section 3 never lists the inputs or outputs for training. It took a long time to figure out that this section describes training and not test time. How are the parameters of the h_s and h_c networks determined?  What feature space are the first two terms in Eq 4; in pixel space?  Eqs 3 and 4 use ||_||_2 notation; does this mean there are square-roots in these terms?

The motivation and explanation of the algorithm coudl also use a lot of improvement.  The Englihs is not very good.
* L26: "existing style transfer methods often confine themselves to the internal style statistics of a single image". This is obviously false, e.g., see pix2pix, CycleGAN, CUTG.
* L31: "... more consistent with human perception" This is a bold claim, either remove it provide citations to justify whatever assertion is being made here.
* L50: "... this is the first work successfully leveraging the power of contrastive learning in style transfer": This is false; see CUT [34].
* L109: "Existing style transfer methods usually produce unsatisfying stylization results with disharmonious colors..."  So the claim here is that all previous methods are terrible?


The video style transfer results are so weak that I think they should be removed if the paper is accepted.  The frame rate of the results is so low that one cannot judge the temporal coherence of the provided videos. These are more like sequences of independent still images.

The paper fails to compare to the state-of-the-art in video style transfer, or to cite it:

Ondřej Jamriška, Šárka Sochorová, Ondřej Texler, Michal Lukáč, Jakub Fišer, Jingwan Lu, Eli Shechtman, and Daniel Sýkora
Stylizing Video by Example
In ACM Transactions on Graphics 38(4):107, 2019 (SIGGRAPH 2019, Los Angeles, USA, July 2019), ACM.

Ondřej Texler, David Futschik, Michal Kučera, Ondřej Jamriška, Šárka Sochorová, Menglei Chai, Sergey Tulyakov, and Daniel Sýkora
Interactive Video Stylization Using Few-Shot Patch-Based Training
In ACM Transactions on Graphics 39(4):73, 2020 (SIGGRAPH, August 2020)


**Time Spent Reviewing:**

1

---

> ### Author Response · Authors · 2021-08-09
> **Dear reviewers, thanks for your careful and invaluable comments! We will address your concerns point by point.**
>
> Q1: It's unclear how these improvements are obtained from the contrastive losses.
>
> A1: We would like to clarify that these improvements are obtained not only from the contrastive losses but also from our external learning strategy. Their effects can be elaborated as follows: (1) The effect of the contrastive losses. As shown in Figure 3, existing style transfer methods usually apply similar style patterns among different styles ($e.g.$, the $1^{st}$, $3^{rd}$, and $6^{th}$ columns of SANet), regardless of their specific characteristics. In comparison, our contrastive losses encourage the network to learn more intra-style compact and inter-style distinguishable style features by pulling the multiple stylization embeddings closer to each other when they share the same style, but pushing far away otherwise. In this way, our method focuses more on the specific characteristics of each style, further refining the stylization results. It is analogous for the content part. (2) The effect of the external learning strategy. This strategy enables our model to learn the human-aware style information, which prevents the distorted texture patterns from appearing in the stylization results and improves the image structures implicitly. We will improve of our description in the future version.
>
> Q2: The writing of the paper is very difficult to follow.
>
> A2: We thank the comments of Reviewer JMx5 that “the paper is well written and easy to follow”, but we would like to improve our description in our future version to resolve Reviewer vsGU’s concerns.
>
> Q3: Section 3 never lists the inputs or outputs for training.
>
> A3: As we introduced in Line121-124 and illustrated in Figure 3, the style image $I_s$ and the content image $I_c$ are the inputs of our model, and the stylized image $I_{sc}$ is the output of our model. We will add more explanations to make them more explicit in our future version.
>
> Q4: How are the parameters of the $h_s$ and $h_c$ networks determined?
>
> A4: $h_s$ and $h_c$ are two learnable MLP (Multilayer Perceptron) networks, whose parameters will be updated during the training stage under the constraint of the loss functions.
>
> Q5: (1) What feature space are the first two terms in Eq. 4, in pixel space? (2) What does $\parallel \ \parallel_2$ mean in Eq. 3 and 4?
>
> A5: (1) Yes, the first two terms in Eq. 4 are in pixel space. Eq. 4 calculates the L2 distance both in pixel space (the first two terms) and in feature space (the last two terms). (2) It means the L2 distance.
>
> Q6: Line26 is obviously false, $e.g.$, see pix2pix, CycleGAN, CUTG.
>
> A6: Pix2pix, CycleGAN, and CUTG are all image domain translation methods. Image domain translation and style transfer are two different tasks. In fact, CycleGAN has distinguished these two tasks in its Section 2, the “Neural Style Transfer” paragraph. According to CycleGAN, image domain translation aims to learn the mapping between two image collections (the source image domain and the target image domain), while style transfer aims to learn the mapping between two specific images (the content image and the style image). Another significant difference between these two tasks is that image domain translation can only translate between two different yet content-similar visual domains, such as horses$\leftrightarrow$zebras, cats$\leftrightarrow$tigers, and day scene$\leftrightarrow$night scene. In comparison, the content image and style image in style transfer can be totally different ($e.g.$, the former is a photo of a person and the latter is van Gogh’s The Starry Night).
>
> Q7: Line31 is a bold claim.
>
> A7: It is a practical intuition that human-created artworks are more consistent with human perception than AI-created stylizations. If AI-created stylizations are more consistent with human perception than human-created artworks, existing style transfer methods’ efforts to mimic human-created artworks will be meaningless. The experimental results in Table 1 also demonstrate this claim. We will improve our description to resolve the reviewer’s concerns in our future version.
>
> Q8: Line50 is False. See CUT [34].
>
> A8: CUT is an image domain translation methods, while our method aims at the style transfer problem. Image domain translation and style transfer are two different tasks, as we analyzed in A6. In addition, CUT used contrastive learning to encourage content preservation, while our method used contrastive learning to learn the stylization-to-stylization relations. To summarize, CUT and our method are two different methods used for two different tasks.
>
> Q9: Does the claim in Line109 mean all previous methods are terrible?
>
> A9: No, we just pointed out the limitations of previous methods yet did not deny their effectiveness in other aspects.
>
> Q10: The video style transfer results are weak, and the frame rate of the results is low.
>
> A10: The heat maps in Figure 4 and the LPIPS distance in Table 2 have quantitatively validated the superior stability and consistency of our method in video style transfer. As for the frame rate, we leveraged the past experiences of LST [24], which used Frame 1 and 5 to exhibit the video style transfer results (Figure 8). Similar to LST [24], we used Frame 21, 25, 28, and 31 in Figure 4, and used Frame 8 and 13 in Figure 5 (b). If the reviewer still argues that the frame rate is low, we would like to use higher frame rate in our future version to address the reviewer’s concerns.
>
> Q11: The paper fails to compare to [A*] and [B*] in video style transfer.
>
> A11: We did not compare with [A*] and [B*] mainly because their goals are too different from ours. Specifically, [A*] focuses on the artistically controlled stylization of video. [B*] aims at the keyframe-based stylization of arbitrary videos, which allows an artist to propagate the style from a few selected keyframes to the rest of the sequence. In comparison, the goal of our method is to promote the stability and consistency for video style transfer. [A*] and [B*] cannot solve our problem, and vice-versa. We will cite these papers and analyze the difference between our method and [A*, B*] in our future version.
>
> [A*] Stylizing Video by Example. SIGGRAPH 2019.
>
> [B*] Interactive Video Stylization Using Few-Shot Patch-Based Training. SIGGRAPH 2020.
>
> Q12: The paper doesn't discuss limitations.
>
> A12: We have discussed the limitations of the proposed method in our supplementary materials.
>
> We sincerely hope that you can raise your score if your concerns are resolved. Thanks!

---

### Official Review · Reviewer_bZMr · 2021-07-14

**Rating:** 7
**Confidence:** 4

**Summary:**

The paper proposes two new additional losses for style transfer:
1. A contrastive loss to incorporate to bring the style of the stylized images with the same style image closer to each other, while pushing the stylized images from different style images further from each other. The same for content.
2. An adversarial loss to learn the general aesthetics of stylized images over a larger dataset.
The authors demonstrate that a combination of these losses plus the regular style-content loss can generate high quality, and surprisingly consistent stylized images, both qualitative and quantitatively as well as user studies.

**Limitations And Societal Impact:**

The broader impact and limitation is missing from the paper and instead is in the supplementary materials. The authors enumerated the positive and negative impacts of their work however it is quite minimal and can be expanded.

**Main Review:**

===== Writing

The paper is very well written but still can be improved.
- The motivation behind the new additions are clearly explained, although it can use more visual examples to demonstrate the shortcomings of the current approaches. Figure 1 can do this with a longer description as why the previous methods fail and how the proposed method shines. In fact, given the subjective nature of style transfer such descriptions are quite important.
- The method is clearly described and is visualized in Figure 2 (texts can be slightly bigger for better readability in print). The reasoning behind the design decisions are clearly described and justified.
- The details of the experiments are noted, however, the training details can be expanded (in supplementary materials). The code is not included and including such details is crucial to attain reproducibility.


===== Methodology

The proposed method is novel, clever and impactful. To the best of my knowledge, and as claimed by the authors, this is the first usage contrastive loss in style transfer and it is applied in a way which is intuitive and makes sense.  The paper tested their proposed method against multiple datasets and the majority of the SOTA models. For qualitative comparison, the authors conducted multiple user studies to capture the deception and preferences score which is pivotal for style transfer papers which lack good reliable metrics.

However, I strongly suggest the authors to include a "non-cherry picked" section of their results to demonstrate the generalization capability of their model. The current set of results, look impressive but cherry-picked. This can be done as part of the supplementary materials or even better a website :)

Another  missing section from the paper, is the training and inference complexity. I suspect by adding the proposed losses the training time substantially increases while the generation runtime stays the same. Training time is important mainly because the proposed model cannot be applied to learning-free style transfer methods.

**Time Spent Reviewing:**

2

---

> ### Author Response · Authors · 2021-08-09
> **Dear reviewers, thanks for your careful and invaluable comments! We will address your concerns point by point.**
>
> Q1: More descriptions of Figure 1 about why the previous methods fail and how the proposed method shines will be better.
>
> A1: Thanks for your suggestion! Previous methods fail because they only focus on increasing the style similarity between the stylization results and the style images, while neglect the harmonization and aesthetics of the stylized image. In comparison, our method can alleviate this issue by leveraging the human-aware style information and the stylization-to-stylization relations. We will add more descriptions in our future version to make our work more persuasive.
>
> Q2: (1) Texts can be slightly bigger for better readability in print. (2) The training details can be expanded. (3) Include a "non-cherry picked" section of the results.
>
> A2: Thanks! We will improve our paper according to your suggestion in our future version.
>
> Q3: How about the training and inference complexity?
>
> A3: Without our proposed losses, the training time is about 21 hours and 40 minutes. With our proposed losses, the training time is about 24 hours and 30 minutes. As for the inference time, it is about 0.06 seconds on resolution 512×512, regardless of with or without our proposed losses. In summary, the proposed losses will increase the training time to some extent, but will not change the inference time. We will add these introductions to our future version.

---

### Official Review · Reviewer_zxcr · 2021-07-15

**Rating:** 3
**Confidence:** 5

**Summary:**

The paper proposes a neural style transfer algorithm, for transferring the artistic style of one image to another.   It is basically SANet [32] with some incremental novelties.

SANet is an encoder-decoder transformer architecture that accepts two images (content/desired style) and outputs a stylized version of the content by optimizing a Gatys-like loss function.

Two novelties are claimed over SANet.

1) the use of ‘human external information’ by which the authors mean a corpus of real artworks in the desired style, to train a GAN discriminator added to the SANet.  It is argued that adding knowledge of multiple real artworks of the desired style in this way helps produce more plausible stylizations.

2) the use of contrastive learning to train their architecture.  The idea here is to ensure pair-wise consistency between style descriptors of generated artwork and the aforementioned corpus of artwork sharing that desired style.


**Ethical Concerns:**

I am not sure this is an ethics consideration but I notice there are non-anonymized funder acknowledgements to some Korean agencies at the end of the paper, isn’t this a violation of NeurIPS submission policy?  I am unsure so flag it here - it did not have any bearing on my review rating.

**Ethics Review Area:**

["Research Integrity Issues (e.g., plagiarism)"]

**Limitations And Societal Impact:**

Societal impact not discussed

**Main Review:**

The main contribution of this paper is extending SANet to draw upon external knowledge – i.e. in addition to taking in a single content/style image pair, a supporting corpus of images with similar content and similar style are provided.
The extensions are fairly straightforward..  the same exact SANet losses are used (ensuring Gatys-like content and style consistency using pre-trained VGG) – here referred to as ‘internal content/style’ losses – and an identity loss agains, as defined, in SANet.

The first technical tweak is adding a loss for the discriminator, which conducts the usual real/fake check on the output of the stylization vs. unpaired set of style images.  This is shown via a ‘deception score’ in Sec.4 - a kind of artistic Turing test user study  -  that the output artwork is considered more realistic, and qualitatively this is shown in the figures of the paper.  The idea is quite simple but seems to improve the output aesthetics.  The practical downside is the need to have a large corpus of images in the desired output style.

The second tweak is adding a contrastive loss and contrastive training methodology to the network – the formation of the batches is explained, but it is not explained how the training curriculum works in tandem with the GAN discriminator.  Is there some alternating training pattern similar to DCGAN?  How is this contrastive training integrated?   A key technical detail omitted from the implementation description is the batch size – it is mentioned that Figure 2 diagram 'uses a batch size b=8 for illustration' but what was used for the experiments reported? Presumably for contrastive learning to make any difference you need batch size in the hundreds or even thousands?  Again, is that practical given the number of style images required – and how would all of this fit into typical GPU VRAM it seems with this architecture it would be challenging.  The actual contrastive loss introduced has limited novelty it is just SimCLR temperature softmax approach with a header network, exactly as in the original SimCLR paper.

Overall I am not convinced by this paper.  The novelty is not significant, and the benefit of the external corpus of style images is not well demonstrated versus the practical disadvantage of requiring it.  The time cost of training such a network to perform a stylization, as well as the undescribed practical issues around large batch size, make me doubt this is a valuable approach.

**Needs Ethics Review:**

Yes

**Time Spent Reviewing:**

1.5

---

> ### Author Response · Authors · 2021-08-09
> **Dear reviewers, thanks for your careful and invaluable comments! We will address your concerns point by point.**
>
> Q1: The practical downside is the need to have a large corpus of images in the desired output style.
>
> A1: We would like to clarify that the datasets used to train our model (MS-COCO: the content dataset; WikiArt: the style dataset) are the same as existing style transfer methods, such as AdaIN [13] and SANet [32]. The difference is that we leveraged the human-aware style information from the style dataset via the GAN loss to enhance the quality of stylizations, while previous methods did not. In addition, similar to existing methods, the content and style datasets are only needed in the training stage. Once trained, our model can generate high-quality stylized images based on arbitrary content and style images, which is very efficient and practical. We will improve our description to avoid misunderstanding in the future version.
>
> Q2: How is this contrastive training integrated? How does the training curriculum work in tandem with the GAN discriminator?
>
> A2: Our network mainly contains a generator and a discriminator. The contrastive losses are combined with other losses ($i.e.$, $L_s$, $L_{adv}$, $L_c$, and $L_{identity}$) introduced in Section 3 to train the generator. As for the discriminator, only $L_{adv}$ is taken to train it. The generator and the discriminator are trained in an alternating manner, just like conventional GANs [9, 35, 2, 51, 3]. We will add more explanations to make them more explicit in our future version.
>
> Q3: Concerns about the batch size.
>
> A3: The batch size in our experiment is set to 16. In this way, for each query, 1 positive example and 13 negative examples are provided, according to the strategy introduced in Section 3.2. Although more negative examples (we denote the number of negative examples as N) usually lead to better performance in contrastive learning, they also result in longer training time and more memory budgets. Therefore, N should be neither too high nor too low. To set an appropriate value for N, we leveraged the past experience of [30, 34]. Specifically, [30] set N to 10, and [34] found their model performed strongly when N was set to 15. Most recently, [A*] considered at most 10 negative samples in the image dehazing task. Therefore, our batch size does not need to be in the hundreds or even thousands. We will add more analysis about the batch size to resolve your concerns in our future version.
>
> [A*] Contrastive Learning for Compact Single Image Dehazing. CVPR 2021.
>
> Q4: The proposed contrastive loss is just the SimCLR temperature softmax approach with a header network. Therefore, the novelty is not significant.
>
> A4: The form of our contrastive loss is indeed inspired by SimCLR. Actually, [30, 34, B*, C*, D*] all adopt the same contrastive loss form as SimCLR. However, as [30] pointed out in their paper, “The most critical design in contrastive learning is how to select the positive and negative pairs”. These methods are still regarded to be very innovative, since they constructed reasonable positive and negative examples for different tasks and solved different problems. Similarly, our contrastive loss is innovative because we adopted an effective strategy to construct the positive and negative pairs for the style transfer task and learned the stylization-to-stylization relation for the first time.
>
> [B*] FSCE: Few-Shot Object Detection via Contrastive Proposal Encoding. CVPR 2021.
>
> [C*] Contrastive Embedding for Generalized Zero-Shot Learning. CVPR 2021.
>
> [D*] Contrastive Learning based Hybrid Networks for Long-Tailed Image Classification. CVPR 2021.
>
> Q5: Concerns about the time cost.
>
> A5: Without our proposed losses, the training time is about 21 hours and 40 minutes. With our proposed losses, the training time is about 24 hours and 30 minutes. As for the inference time, it is about 0.05 seconds on resolution 512×512, regardless of with or without our proposed losses. In summary, the proposed losses will increase the training time to some extent, but will not change the inference time.
>
> Q6: Societal impact not discussed.
>
> A6: We have discussed our societal impact in the supplementary materials.
>
> Q7: There are non-anonymized funder acknowledgements to some Korean agencies at the end of the paper?
>
> A7: We can assure the reviewer that our work has nothing to do with any Korean agencies. As far as we know, we never acknowledged anyone or any agency in our paper. Could the reviewer provide the specific line number of the so-called “non-anonymized funder acknowledgements”? In this way, we can make more detailed explanations.
>
> We sincerely hope that you can raise your score if your concerns are resolved. Thanks!

---

### Official Review · Reviewer_JMx5 · 2021-07-17

**Rating:** 6
**Confidence:** 4

**Summary:**

In this paper, the authors introduce an internal-external learning framework for style transfer. In the proposed method, the classical VGG-based style loss, which encourages the stylized result to capture the low-level statistics of the style image, is combined with a GAN loss designed to capture the style priors from the style database. In addition, contrastive losses are designed to encourage the results to better capture the stylization-to-stylization relation. Experiments in the paper demonstrate the proposed method can generate visually more appealing results compared to previous style transfer methods.

**Limitations And Societal Impact:**


The limitation and societal impacts have been discussed in the paper (incorporated in the supplementary materials).

**Main Review:**


Strength:
 + The paper is well written and easy to follow.
 + The idea of incorporating GAN loss to capture the style priors from a style database is interesting. The provided results seem to indicate that such external loss can help produce results that look more natural compared to previous works.
 + The evaluation processes for evaluating the stylization results and comparison with existing works are solid, with both visual results on a range of different content and a thorough user study.

Weaknesses:
 + I found the use of the term "internal learning" in this context a bit misleading. Internal learning has been used in the literature to refer to the process of optimizing the model parameters on a single image at **test time**. In this paper, the internal learning part seems to refer to the conventional style-transfer loss (which consists of VGG-based losses for style and content loss), which is used at training time and applied when trained on the whole dataset rather than for test-time optimization. I think it is fine to overload the term if it serves the purpose but clarification needs to be provided to avoid confusion.
 + While I find it interesting to explore applying GAN loss to style transfer problem and I appreciate the improvement it brings to the results, I feel that more discussions and examples should be provided to analyze the effects of it. In particular, one challenge in incorporating GAN loss into this setting could happen when the provided style image has a unique style that does not appear in the style database. How would (and how should) the model behave in that case? I would appreciate more discussions on that regard along with more examples to analyze the benefit of incorporating GAN loss.
 + It's also not clear how much improvement was gained by the incorporation of the contrastive loss. Only three examples were provided (in Fig. 5) and the improvements from using contrastive loss in those examples do not seem obvious to me. More explanations on this would make the contributions of these additional losses better justified.


**Time Spent Reviewing:**

3

---

> ### Author Response · Authors · 2021-08-09
> **Dear reviewers, thanks for your careful and invaluable comments! We will address your concerns point by point.**
>
> Q1: Concerns about the term “internal learning”.
>
> A1: Thanks for your suggestion! We will improve our description and clarify this term to avoid confusion in our future version.
>
> Q2: What if the provided style image has a unique style that does not appear in the style dataset?
>
> A2: In the training stage, the provided style image $I_s$ always comes from the style dataset $S$ ($i.e.$, $I_s \in S$), according to the definition of our task in Line121-124. After trained on the large-scale style dataset $S$, our model learned the generalized human-aware style information via the GAN loss. Therefore, in the inference stage, the provided style image can be arbitrary, just like [5, 13, 26, 37, 24]. In fact, none of the style images in Figure 3 have appeared in the style dataset $S$, demonstrating the generalization ability of our method.
>
> Q3: More discussions and examples should be provided to analyze the effects of the GAN loss and contrastive loss.
>
> A3: Thanks for your suggestion! In our current version, we provided some ablation studies in Section 4.4 to analyze the effects of the GAN loss and contrastive loss. To make our work more solid, we would like to add more quantitative and qualitative results, and provide more insightful analysis of the effects in our future version: (1) The effect of the GAN loss. As shown in Figure 1, the stylized images generated by existing style transfer methods usually contain some disharmonious colors and repetitive patterns, which makes them easily distinguishable from real artworks. This is because these methods only focus on increasing the style similarity between the stylized image and the style image. In comparison, equipped with the GAN loss, our model removes the distorted texture patterns and better preserves the image structures, so that the discriminator of GAN cannot distinguish the synthesized stylizations from the real artworks after the adversarial training. (2) The effect of the contrastive losses. As shown in Figure 3, existing style transfer methods usually apply similar style patterns among different styles ($e.g.$, the $1^{st}$, $3^{rd}$, and $6^{th}$ columns of SANet), regardless of their specific characteristics. In comparison, our contrastive losses encourage the network to learn more intra-style compact and inter-style distinguishable style features by pulling the multiple stylization embeddings closer to each other when they share the same style, but pushing far away otherwise. In this way, our method focuses more on the specific characteristics of each style, further refining the stylization results. It is analogous for the content part.
>
> We sincerely hope that you can raise your score if your concerns are resolved. Thanks!

---

### Review · Ethics_Reviewer_YEHD · 2021-07-22

**Recommendation:**

I'm not sure what the reviewer is referring to. I looked at both the main submission and the supplement and didn't find any acknowledgement of funding sources. As far as I can tell, the issues flagged just doesn't apply to this paper.

**Ethics Review:**

One of the reviewers flagged the paper for violating anonymization guidelines by including funding acknowledgements.

---

### Review · Ethics_Reviewer_D5WN · 2021-08-13

**Recommendation:** N/a

**Ethics Review:**

The contribution of the work aims to improve style transfer in imagery like artwork. The ethical ramifications of this work (e.g. injury, safety, security, human rights, surveillance) are not very salient.

---

### Decision · Program_Chairs · 2021-09-27

**Decision:**

Accept (Poster)

**Comment:**

The submission proposes an extension of SANet's artistic style transfer loss formulation which adds adversarial and contrastive terms to the loss. The adversarial loss term is computed using a discriminator trained to classify images as either belonging to the style dataset or to the distribution of stylized content images. The contrastive loss terms encourage the embeddings of stylized images which either share the same content or style image to be closer and vice versa.

The proposed approach is compared against SANet and other baselines through qualitative (showing stylizations) and quantitative (user study, LPIPS) means. Stylizations are compared for a few content/style image pairs to qualitatively assess the effect of the adversarial and contrastive loss terms.

Reviewers appreciate the writing quality and agree on the fact that the results presented in the submission represent an improvement over previous approaches, but there is disagreement on whether the analysis of the effect of individual loss components is sufficient, and on whether the limitations of the GAN loss are sufficiently discussed.

On the limitations of the GAN loss, reviewer JMx5 objects that its benefit may not be realized for style images that are very different from the training styles. The authors respond that the style images they evaluate on have not been seen during training. Reviewer zxcr points out in the reviewer discussion that the extent to which the effect of the GAN loss generalizes to held-out style images likely depends on whether they are in-distribution or out-of-distribution (the "distribution" here being the distribution of style images seen during training). This is a good point, and the authors' response here isn't entirely satisfactory: even if the model generalizes to in-distribution held-out style images, there's no guarantee that it would generalize to out-of-distribution style images, and that limitation should be discussed. In the discussion, Reviewer vsGU mentions that discussing the limitations of the proposed approach belongs to the main paper and not the Appendix. I agree with that.

There was also a debate on whether the distinction between style transfer and domain translation is artificial, which I think requires some nuance. Since domain translation approaches work by imitating the statistics of a population of domain-specific images, it does feel weird that the submission dismisses style transfer approaches as "neglecting the external style information reserved in the large-scale style dataset" without acknowledging domain translation approaches. There is a distinction between the two (the former targets a specific style image while the latter targets the common "look and feel" of a collection of style images), but I think the submission would benefit from being more generous in drawing connections between various families of approaches.

On the influence of the added loss terms, I would break the issue down into two questions:
- Are the current ablations convincing enough?
- If they are not, how much does that weaken the submission?

I agree with Reviewers JMx5 and zxcr that a qualitative assessment on three content/style image pairs is a very small number of data points to draw conclusions from. This issue could have been fixed by adding ablated models among the approaches evaluated in Section 4.3's user study.

On whether this is a serious issue, the risk here is that the effect of one of the two loss terms could be marginal, which would nullify one of the submission's two major contributions. The principle of incorporating global information from a corpus of styles into the training signal remains an interesting contribution nevertheless, but obviously the paper's impact would be more substantial if it better characterized the contributions of the two loss terms.

This submission is very much a borderline case. I read the paper attentively, and I agree that the idea of incorporating information from a corpus of styles rather than treating each style as separate is a new and interesting contribution which incorporates insights from domain translation. Even though the submission has its flaws, I recommend accepting it provided the following is addressed in the final manuscript:
- A discussion on the approach's limitations, and in particular those of its GAN loss, is incorporated into the main text.
- The relationship between the proposed approach and domain translation is better highlighted.
- Additional examples are shown in the Appendix to accompany Figure 5.